# Endothelial Dysfunction in Primary Sjögren’s Syndrome: Correlation with Serum Biomarkers of Disease Activity

**DOI:** 10.3390/ijms241813918

**Published:** 2023-09-10

**Authors:** Alexandru Caraba, Stela Iurciuc, Mihaela Nicolin, Mircea Iurciuc

**Affiliations:** 13rd Internal Medicine, Diabetes and Rheumatology Department, University of Medicine and Pharmacy “Victor Babeș”, 300041 Timișoara, Romania; alexcaraba@yahoo.com; 2Cardiology Department, University of Medicine and Pharmacy “Victor Babeș”, 300041 Timișoara, Romania; mirceaiurciuc@gmail.com; 3Cardiology Department, “Victor Popescu” Military Hospital, 300080 Timișoara, Romania; nicolinmihaela@yahoo.com

**Keywords:** beta-2 microglobulin, cytokines, endothelial dysfunction, Sjögren’s syndrome

## Abstract

To assess the relationship between endothelial dysfunction and serum cytokines, anti-SSA and anti-SSB antibodies, beta-2 microglobulin levels, focus score and EULAR Sjögren’s Syndrome Disease Activity Index (ESSDAI) in primary Sjögren’s syndrome (pSS) patients. The study included 90 patients with pSS and 45 healthy subjects, matched for age and gender, as controls. Serum beta-2 microglobulin, total cholesterol, HDL-cholesterol, triglycerides, TNF-α, and IL-6 were analyzed in both the groups. Patients with pSS were also tested for antinuclear antibodies, anti-SAA (anti-Sjögren’s syndrome-related antigen A) antibodies, anti-SSB (anti-Sjögren syndrome related antigen B) antibodies, and focus score (the histopathologic one, based on minor salivary gland biopsy). Endothelial dysfunction was assessed by means of flow-mediated dilation (FMD) in the brachial artery. Data are presented as mean ± standard deviation. Statistical analysis was performed using the *t*-test and the Pearson’s correlation. Differences were considered to be statistically significant if the value of *p* < 0.05. Endothelial dysfunction was identified in pSS patients (*p* < 0.00001). The serum levels of cytokines (TNF-α, respective IL-6) and beta-2 microglobulin were increased in pSS patients compared with controls (*p* < 0.00001). Endothelial dysfunction (expressed as FMD%) was correlated with focus score, ESSDAI, levels of anti-SSA and anti-SSB antibodies, beta-2 microglobulin, IL-6, and TNF-α, with statistical significance. Endothelial dysfunction is present in pSS patients and is associated with a high focus score and activity as well as increased concentrations of antibodies, pro-inflammatory cytokines, and beta 2-microglobulin.

## 1. Introduction

Sjögren’s syndrome (SS) is an autoimmune disease, characterized by the sicca complex [1]. It affects mainly middle-aged women, being primary (pSS) or secondary (sSS). Secondary SS is found to be associated with other rheumatic diseases, such as rheumatoid arthritis, systemic lupus erythematosus, systemic sclerosis, dermatomyositis, mixed connective tissue disease, etc. The clinical features are related to lymphocytic infiltration of exocrine glands, mainly salivary and lacrimal glands. In addition to exocrine glands, lymphocytic infiltration may involve other organs and systems, such as lungs, gastrointestinal tract, kidneys, vessels, and central or peripheral nervous system. About 30–40% of pSS patients present with systemic involvement [2,3]. pSS activity is evaluated through the EULAR Sjögren’s Syndrome Disease Activity Index (ESSDAI) [4].

In 2016, ACR-EULAR (American College of Rheumatology–European League Against Rheumatism) validated the classification criteria of pSS, one of these being the histological criterion, which requires biopsy of the minor salivary glands and calculation of the focus score. This represents the density of foci (tight clumps of lymphocytes ≥ 50) on a surface of 4 mm^2^. A focus score ≥ 1 is considered to be positive [5,6]. 

Environmental factors (various infections, vitamin D, estrogen, stress, smoking, silicone implants) acting on a genetic predisposition contribute to the activation of T and B lymphocytes, cytokine production, autoantibody generation by activated B lymphocytes, and the formation of glandular germinal centers, finally resulting in destruction of the salivary and lacrimal gland epithelial cells [7,8,9]. pSS patients have high levels of cytokines produced by Th1, Th2, Th17, Tregs, B cells and salivary gland epithelial cells, including IL-1, IL-2, IL-4, IL-5, IL-6, IL-17, IL-21, IL-22, TNF-α, IFN-γ, macrophage colony-stimulating factor (M-CSF), B cell activating factor (BAFF), and mast cell growth factor (MCGF) [10,11,12]. Furthermore, studies have documented that some of these cytokines are related to the atherosclerosis process [13].

Beta-2 microglobulin, a 11.8 kDa protein comprises 100 amino acids and is considered to be an important component of the major histocompatibility complex class I (MHC-I) molecule. It is expressed on the surface of almost all nucleated cells [14]. High levels of circulating beta-2 microglobulin are associated with chronic inflammatory conditions and impaired renal function [14,15]. Ho et al. stated that high levels of serum beta-2 microglobulin are associated with high cardiovascular risk in the general population [16].

Atherosclerosis is a multifaceted process, which begins early in life, having a subclinical evolution, while the manifestations appear later in life. The presence of lymphocytes and macrophages in atherosclerotic plaques indicates that the immune system is involved in pathogenesis of this condition [17]. Atherosclerosis associated with rheumatic inflammatory diseases is considered to be the result of interactions between the traditional cardiovascular risk factors, chronic inflammation, immunological abnormalities, and some drug therapies that damage the vascular endothelium [18]. Atherosclerosis related to pSS is less studied. Atherosclerotic cardiovascular events in pSS patients are reported to be about 61.6%, versus 29.7% in healthy controls [19]. Atherogenesis is a continuous process, developed over several years. During its evolution, endothelial dysfunction represents the first step [17]. Endothelial dysfunction is evaluated by means of flow-mediated dilation (FMD) in the brachial artery [20].

The involvement of cytokines, such as IL1 [21], IL6 [22], TNF-α [23], IFN-γ [24], IL-10 [25], and IL-22 [26] in cardiovascular atherosclerotic events is well known. These cytokines are considered important determinants for atherosclerosis, regardless of other atherosclerosis risk factors [27]. 

The aim of this study was the assessment of the relationship between endothelial dysfunction and serum cytokines, anti-SSA and anti-SSB antibodies, beta-2 microglobulin levels, focus score, and EULAR Sjögren’s Syndrome Disease Activity Index (ESSDAI) in pSS patients.

## 2. Results

Baseline demographic data of pSS patients and controls are presented in Table 1.

Schirmer’s test was positive in all the pSS patients, with mean values of 2.25 ± 1.83 mm, whereas none of the controls had a positive Schirmer’s test. Focus score had mean values of 4.64 ± 1.55. Activity of pSS was evaluated by means of ESSDAI. The mean value of ESSDAI was 14.78 ± 7.07. Based on the ESSDAI value, the pSS patients were classified as low activity level (22 patients; ESSDAI: 3.04 ± 0.72), medium activity level (38 patients; ESSDAI: 8.54 ± 3.10), and high activity level (30 patients; ESSDAI: 21 ± 4.20). 

Antinuclear antibodies, anti-SSA (anti-Sjögren’s syndrome-related antigen A) and anti-SSB (anti-Sjögren’s syndrome-related antigen B) antibodies, and rheumatoid factor were identified in all pSS patients. The mean values of these parameters were 79.57 ± 60.86 u/mL (anti-SSA antibodies), 60.38 ± 58.45 u/mL (anti-SSB antibodies), and 125.66 ± 72.03 u/dL (rheumatoid factor). 

pSS patients were found to have lower leucocyte and lymphocyte counts in comparison to the subjects from the control groups, with statistical significance (*p* < 0.00001). 

Endothelial dysfunction was noted in patients suffering from pSS, it was assessed by means of FMD. The FMD value in these patients was about 9.19 ± 2.15%, compared to 14.06 ± 1.88% in controls, with statistical significance (*p* < 0.00001). No gender differences in FMD values were identified in pSS patients (10.32 ± 2.48% in females versus 9.82 ± 2.45% in males, *p* > 0.05) or in controls (13.89 ± 1.70% in females versus 14.49 ± 2.01% in males, *p* > 0.05).

The serum levels of cytokines (TNF-α, respective IL-6) was higher in pSS patients than in controls, and this was statistically significant (*p* < 0.0001 and *p* < 0.00001, respectively). Likewise, beta-2 microglobulin was increased in pSS patients more than in the control group (*p* < 0.00001). From a statistical point of view, no significant differences were noted among the groups, in terms of total cholesterol, HDL-cholesterol, triglycerides, or systolic and diastolic blood pressure. These findings are presented in Table 2.

Glandular inflammatory infiltrate (assessed by means of the focus score) was associated with increased serum levels of antibodies, proinflammatory cytokines, and beta-2 microglobulin, generating endothelial dysfunction. In contrast, focus score was found to be associated with reduced leucocyte and lymphocyte counts. 

Focus score was correlated with the levels of antibodies and levels of circulating cytokines, serum beta-2 microglobulin, and leucocyte and lymphocyte counts, as shown in Table 3.

The present study revealed a correlation with statistical significance between focus score and pSS activity (ESSDAI) (r = 0.9603, *p* < 0.00001).

The correlations identified between the endothelial dysfunction (expressed as FMD%) and focus score, pSS activity (ESSDAI), levels of anti-SSA and anti-SSB antibodies, beta-2 microglobulin, IL-6, and TNF-α are presented in Table 4. 

## 3. Discussion

Sjögren’s syndrome (SS) is a chronic, systemic autoimmune exocrinopathy, which mainly affects the lacrimal and salivary glands, resulting in their dysfunction with the onset of sicca manifestations [13]. SS affects about 0.1–4% of the population, especially females (female to male ratio 9/1) in their fifth decade of life [28]. Sjogren’s syndrome can be primary (pSS) or secondary (sSS), due to other underlying diseases, most commonly rheumatoid arthritis or systemic lupus erythematosus [3]. In addition to glandular involvement, pSS also has extraglandular manifestations. Endothelial dysfunction, the first step in atherosclerosis development, has been identified in pSS patients [29,30,31]. 

Endothelial dysfunction was identified by means of FMD in the brachial artery. The FMD value < 11.11% had a sensitivity of 80%, a specificity 86.67%, a positive predictive value of 76.66%, and a negative predictive value of 83.33% for cardiovascular atherosclerotic disease [32]. On the other hand, FMD of >6.5% excluded coronary artery disease (95% sensitivity, 60% specificity) and FMD < 3.1% excluded 95% of healthy individuals (95% specificity, 31% sensitivity) [33]. 

The present study revealed the presence of endothelial dysfunction in pSS patients (FMD: 9.19 ± 2.15% in pSS patients versus 14.06 ± 1.88 in controls, *p* < 0.00001). Many authors have reported similar findings, confirming the links between endothelial dysfunction and pSS. In one study, Pirildar et al. demonstrated that FMD was impaired in 25 pSS patients compared with the 29 healthy subjects, which were considered to be a control group, with statistical significance (*p* < 0.001) [34]. Akyel et al. showed impaired flow mediated dilation in 35 pSS patients, the results being statistically significant (7% vs. 12%, *p* = 0.002) [35]. Similar results were confirmed by Yong et al. [3] and Gerli et al. in their studies, suggesting that a functional impairment of the arterial wall may support the early phases of atherosclerotic damage in pSS [36].

In a state of health, vascular endothelium produces vasodilators (nitric oxide, prostacyclin 2), anticoagulant mediators (thrombomodulin), and fibrinolytic substances (tissue plasminogen activator). However, during chronic inflammation, as seen in pSS, endothelium is activated, becoming dysfunctional, which is known as endothelial dysfunction. Endothelial dysfunction is associated with overproduction of prothrombotic mediators, overexpression of adhesion molecules and, furthermore, it reduces the release of vasodilatory factors [37,38]. Valim et al. suggested that proinflammatory cytokines, along with the presence of an imbalance between endothelial damage and repair, should be considered to be the most important factors involved in endothelial dysfunction associated with pSS [39]. 

Impairment of endothelial function was more pronounced in active pSS, showing an inverse strong correlation between FMD and ESSDAI (r = −0.9456, *p* < 0.00001). On the other hand, a correlation was found between the endothelial dysfunction and the focus score (r = −0.9029, *p* < 0.00001). In a study performed by Luczak et al. on 46 patients with pSS and 30 control subjects without known cardiovascular disease revealed that the FMD was correlated with disease activity (*p* = 0.02), focus score (*p* = 0.04), anti SS-A-antibody titer (*p* = 0.03), and pulmonary involvement (*p* = 0.001) [40]. Similar results were reported by Gerli et al. [36], Alunno et al. [41] and Pasoto et al. [42]. 

All our investigated pSS patients presented antinuclear, anti-SSA and anti-SSB antibodies, showing a negative correlation between these parameters and FMD. However, the titer of these antibodies was correlated with the focus score as well (Table 4). Similar results were reported by Gerli et al., demonstrating a significant correlation between endothelial dysfunction and titer of anti-SSB antibodies (*p* = 0.02) [36]. Vaudo et al. showed that the pSS patients with antinuclear antibodies, anti-SSA/Ro antibodies, or anti-SSA/Ro plus anti-SSB/La antibodies presented higher focus scores than the patients without these autoantibodies (*p* < 0.01). Focus score was significantly greater in patients with serum anti-SSA/Ro plus anti-SSB/La antibodies in comparison to the patients having anti-SSA/Ro antibodies alone (*p* < 0.05) [43].

Other laboratory findings associated with focus score were leucopenia and lymphopenia, having significant correlations (r = −0.8519, *p* < 0.00001 and r = −0.7587, *p* < 0.00001, respectively). Vaudo et al. identified a significant correlation between focus score and leucopenia (*p* < 0.02) [43].

On the other hand, the existence of a correlation between the titer of these antibodies and the focus score (measure of the lymphocytic infiltration), is still unclear, raising questions regarding the existence of a similar process in the subendothelial space. Endothelial dysfunction was more evident in pSS patients that presented with severe glandular enlargement, articular involvement, and positive anti-SSA antibodies [36,44].

TNF-α and IL-6 are involved in the appearance of endothelial dysfunction [45]. In pSS, IL6 is more important than TNF-α as it is related to the occurrence of specific manifestations found during the course of the disease, including even the cardiovascular ones [46]. Our study showed negative correlations between the level of these cytokines and FMD (IL-6: r = −0.8931, *p* < 0.00001, TNF-α: r = −0.6597, *p* < 0.0001). As can be seen, the correlation is stronger in the case of IL-6 than in the case of TNF-α. Weiner et al. reported a significant inverse correlation between IL-6 levels and FMD (r = −0.042; *p* = 0.02) after adjustment for age, gender, race/ethnicity, education, income, low-density lipoprotein, diabetes, glucose, hypertension status and treatment, waist circumference, triglycerides, baseline brachial diameter, recent infection, and the use of medications that may alter inflammation [47]. In pSS pathogenesis, TNF-α is not particularly important, this fact being proven by the lack of efficiency of anti-TNF-α therapy in these patients [45,48,49]. 

Our study showed elevated levels of beta-2 microglobulin in pSS patients compared with controls, with statistical significance (3.78 ± 1.03 mg/L vs. 2.02 ± 0.48 mg/L, *p* < 0.00001). These levels were correlated with focus score (r = 0.9455, *p* < 0.00001), pSS activity (r = 0.9390, *p* < 0.00001) and with impaired endothelial function (r = −0.8709, *p* < 0.00001). Beta-2 microglobulin is found in large quantities on the surface of lymphocytes and monocytes, its synthesis is regulated by interferons and the proinflammatory cytokines. In 1978, Strom et al. revealed that the patients with SS have elevated serum beta-2 microglobulin levels, higher mean age, and elevated antinuclear antibodies. Elevated serum beta-2 microglobulin was also found to be associated with a higher incidence of complications, including cardiovascular complications [50]. High levels of beta-2 microglobulin is associated with a high risk of cardiovasular disease through endothelial dysfunction [51]. Beta-2 microglobulin is a biomarker that might be associated both with vascular inflammation and its dysfunction [52]. In active pSS, with a high value of ESSDAI, an important glandular inflammatory infiltrate, measurable by means of focus score, consisting of active lymphocytes B, T1, T2, and T17, is associated with high levels serum beta-2 microglobulin and anti-SSA and anti-SSB antibodies [53]. Gottenberg et al. reported in their study that pSS patients showed high levels of beta-2 microglobulin. In contrast, the patients with elevated beta-2 microglobulin had higher ESSDAI scores (*p* < 0.0001) [54]. In a study conducted on 81 patients with pSS, Tecer et al. reported that the serum beta-2 microglobulin levels are significantly higher in patients with anti-SSA and anti-SSB antibodies compared to patients positive for anti-SSA only and patients negative for both antibodies (*p* < 0.001). Serum beta-2 microglobulin level was significantly correlated with ESSDAI (r = 0.482, *p* = 0.001) [55]. Julich et al. reported similar findings in regard to the correlation between the serum concentrations of beta-2 microglobulin and ESSDAI (r = 0.421, *p* = 0.004) [56]. 

Until now, there have not been many studies related to endothelial dysfunction in pSS. Different aspects related to endothelial dysfunction in pSS have been described, as revealed in the Section 3. This study aimed to unite all aspects related to endothelial dysfunction and pSS activity markers.

This study has certain limitations. The studied patients and control group individuals were enrolled from the southwestern region of Romania. All of them were Caucasians. Moreover, this study investigated the contribution of only some cytokines (TNF-α, IL-6) in the development of endothelial dysfunction, but a much higher number of cytokines are increased in pSS. Future studies are required to evaluate the contribution of other cytokines to the occurrence of endothelial dysfunction.

## 4. Material and Methods

### 4.1. Patients

A total of 90 consecutive patients with pSS were enrolled in this case-control study, which was performed in the Internal Medicine Department, Timișoara, Romania between July 2019 and July 2023. In all, 45 healthy subjects, matched for age and gender, were enrolled as controls. All patients fulfilled the 2016 ACR/EULAR Classification Criteria for pSS [5]. Exclusion criteria were as follows: age under 18 years, inflammatory diseases in 30 days before the investigation, sSS, overlap syndromes, sicca symptoms related to sarcoidosis, IgG4-related disease, hepatitis C infection, acquired immunodeficiency syndrome, previous head and neck radiation therapy, graft versus host disease, uncontrolled diabetes mellitus, amyloidosis, chronic kidney disease (eRFG < 60 mL/min/1.73 m^2^), pregnancy or breastfeeding, current use of drugs that might decrease salivary gland function, and current smoking. The control group comprised healthy attendants coming to the Internal Medicine Department with the pSS patients. All the patients and controls gave their written informed consent prior to enrollment in the study. The study was approved by the Ethics Committee of the Railway Clinical Hospital Timișoara, Romania, with registration number 139/July 2019. The study was in accordance with the Declaration of Helsinki.

### 4.2. Methods

Medical history and current medication charts were noted in both the pSS patient group and the control group and complete clinical examination was performed. The disease activity was evaluated using EULAR Sjögren’s Syndrome Disease Activity Index (ESSDAI), which includes 12 domains: constitutional, lymphadenopathy, glandular, articular, cutaneous, pulmonary, renal, muscular, peripheral nervous system, central nervous system, hematologic, and biologic; values < 5 represented low disease activity, 5–13 moderate disease activity, and over 14 high disease activity [4].

Antinuclear, anti-SSA and anti-SSB antibodies, rheumatoid factor, serum beta-2 microglobulin, total cholesterol, HDL-cholesterol, triglycerides, TNF-α, IL-6, leucocytes, and lymphocyte were measured in all pSS patients. Serum beta-2 microglobulin, total cholesterol, HDL-cholesterol, triglycerides, TNF-α, IL-6, leucocytes, and lymphocytes were measured in all controls.

Antinuclear antibodies were detected by indirect immunofluorescence (HELMED), anti-SSA and anti-SSB antibodies were determined using fluoroimmunoenzymatic assay. Rheumatoid factor was detected by latex agglutination test. Serum beta-2 microglobulin was performed using immuno-enzymometric assay with chemiluminescence detection (CLIA-serum, limit of detection 24.7 pg/mL). Total cholesterol, HDL-cholesterol, and triglycerides were determined using the serum spectrophotometric method. Cytokines (TNF-α and IL-6) were detected using the chemiluminescence method (CLIA-serum limit of detection 1.7 pg/mL and electrochemiluminescence method (ECLIA-serum, limit of detection 0.04 pg/mL), respectively.

For focus score assessment, minor salivary gland biopsy was carried out in all the pSS patients. The minor salivary glands are very accessible and used for calculating the focus score. They are located under the inner surface of the lip. After numbing with Lidocain 10%, an incision was made on the inner surface of the lip, removing 5–7 glands. Using hematoxilin-eosin stain, the pathologist identified tight clumps of lymphocytes (≥50), called foci. Their density on a surface of 4 mm^2^ defined the focus score. A focus score ≥ 1 was considered to be positive [6]. 

Endothelial dysfunction was assessed by means of flow-mediated vasodilation (FMD) in the brachial artery, using B-mode ultrasonography (Siemens Acuson X300 Ultrasound System, with linear transducer of 10 MHz, Siemens, Munich, Germany). Before the test, the patient rested at a stable room temperature between 20–25 °C; smoking and ingestion of caffeine, high-fat foods, and vitamin C were prohibited. The diameter of the brachial artery was measured incidentally with the R wave of the electrocardiograph trace (Di). Then, ischemia was induced by inflating the pneumatic cuff to a pressure 50 mmHg above systolic pressure in order to obliterate the brachial artery and induce ischaemia. After 5 min, the cuff was deflated, and the diameter was measured at 60 s post deflation (Df). FMD was calculated using the formula [(Df − Di)/Di] × 100 [20].

### 4.3. Statistical Analysis

The normal distribution of data was assessed using the Kolmogorov–Smirnov test. Normally distributed data were reported as mean ± standard deviation. Statistical analyses were performed using parametric tests: *t*-test and Pearson’s correlation. Differences were considered statistically significant if the value of *p* was < 0.05. 

## 5. Conclusions

Endothelial dysfunction is present in pSS patients. Rich lymphocytic inflammatory infiltrate is associated with high focus scores, high pSS activity, increased concentrations of antibodies, pro-inflammatory cytokines, beta 2-microglobulin, and impairment of endothelial function.

## Figures and Tables

**Table 1 ijms-24-13918-t001:** Demographic data in pSS patients and controls.

Parameter	Value (Mean ± Standard Deviation)	
	pSS Patients	Controls
Sex [n (%)] Males Females	90 30 (33.33%) 60 (66.66%)	45 16 (35.55%) 29 (64.44%)
Mean age (years)	55.78 ± 7.09	54.73 ± 5.76
Mean length of pSS evolution (years)	6.92 ± 3.78	-
Immunosuppressive drugs used by the pSS patients at the time of investigation	Hydroxychloroquine (26 patients) Azathioprine (19 patients) Methotrexate (16 patients) Mycophenolate mofetil (15 patients)	-
Extra-glandular involvement in pSS patients		-
Malignant lymphoma	3 patients (3.33%)	
Articular involvement	25 patients (27.22%)	
Pulmonary involvement	20 patients (22.22%)	
Renal involvement (only renal tubular acidosis type 1)	9 patients (10%)	
Cutaneous vasculitis	10 patients (11.11%)	

**Table 2 ijms-24-13918-t002:** Monitored parameters assessed in pSS patients and controls.

Parameter	pSS Patients	Controls	*p*
FMD (%)	9.19 ± 2.15	14.10 ± 1.82	<0.00001
TNF-α (pg/mL)	21.85 ± 14.43	8.32 ± 5.75	<0.0001
IL-6 (pg/mL)	36.75 ± 17.58	9.32 ± 10.97	<0.00001
Beta-2microglobulin (mg/L)	3.78 ± 1.03	1.98 ± 0.48	<0.00001
Anti-SSA antibodies (u/mL)	79.57 ± 60.86	-	
Anti-SSB antibodies (u/mL)	60.38 ± 58.45	-	
Leucocyte count (per microliter)	3132.57 ± 1174.66	7495.75 ± 1244.81	<0.00001
Lymphocyte count (per microliter)	723.51 ± 377.02	2260.55 ± 420.98	<0.00001
Total cholesterol (mg/dL)	178.2 ± 37.97	192.12 ± 23.12	NS
HDL-cholesterol (mg/dL)	50.57 ± 8.98	53.72 ± 8.56	NS
Triglycerides (mg/dL)	156.16 ± 46.45	148.47 ± 52.25	NS
Systolic BP (mmHg)	141.09 ± 21.07	145.62 ± 22.82	NS
Diastolic BP (mmHg)	85.54 ± 12.21	82.41 ± 18.62	NS

(NS: not significant).

**Table 3 ijms-24-13918-t003:** Correlations between focus score and monitored parameters.

Correlation between Focus Score and	r	*p*
Anti-SSA antibodies	0.9314	<0.00001
Anti-SSB antibodies	0.2783	<0.01
Beta-2 microglobulin	0.9455	<0.00001
IL-6	0.9553	<0.00001
TNF-α	0.5796	<0.0001
Leucocyte count	−0.8519	<0.00001
Lymphocyte count	−0.7587	<0.00001

**Table 4 ijms-24-13918-t004:** Correlations between FMD and monitored parameters.

Correlation between FMD and	r	*p*
Focus score	−0.9029	<0.00001
ESSDAI	−0.9456	<0.00001
Anti-SSA antibodies	−0.9682	<0.00001
Anti-SSB antibodies	−0.4071	<0.001
Beta-2 microglobulin	−0.8709	<0.00001
IL 6	−0.8931	<0.00001
TNF-α	−0.6597	<0.0001

## Data Availability

Not applicable to this article.

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
