# Peer review of "Endothelial Dysfunction in Primary Sjögren’s Syndrome: Correlation with Serum Biomarkers of Disease Activity"

_ijms, 2023, doi:10.3390/ijms241813918_

Round 1
Reviewer 1 Report
The paper on “Endothelial dysfunction in primary Sjögren’s syndrome: correlation with serum biomarkers of activity” reported on interesting research paper regarding the pathophysiology of Sjögren’s Syndrome, as well as . The study was well designed. Also, the authors used a good range of modern technologies to corroborate their aim. This is an interesting paper on an interesting topic. The paper is nicely written and acceptable. Although further studies are need to clarify the exact correlation between the endothelial dysfunction and disease activities (serum biomarkers of activit) in Sjögren’s syndrome , their article represents a landmark work in this field and is worth publishing as "communications" in the journal.
Author Response
Dear Reviewer,
Thank you for taking the time to review the manuscript and for your appreciation.
Thank you,
Best regards,
Alexandru Caraba

Reviewer 2 Report
The study is quite novel although these references should be quoted as they mark what is really new:
Ren Y, Cui G, Gao Y. Research progress on inflammatory mechanism of primary Sjögren syndrome. Zhejiang Da Xue Xue Bao Yi Xue Ban. 2021 Dec 25;50(6):783-794. English. doi: 10.3724/zdxbyxb-2021-0072.
Pirildar T, Tikiz C, Ozkaya S, Tarhan S, Utük O, Tikiz H, Tezcan UK. Endothelial dysfunction in patients with primary Sjögren's syndrome. Rheumatol Int. 2005 Sep;25(7):536-9. doi: 10.1007/s00296-005-0599-5.
Łuczak A, Małecki R, Kulus M, Madej M, Szahidewicz-Krupska E, Doroszko A. Cardiovascular Risk and Endothelial Dysfunction in Primary Sjogren Syndrome Is Related to the Disease Activity. Nutrients. 2021 Jun 17;13(6):2072. doi: 10.3390/nu13062072.
Methods and statistical issues need relevant changes as detailed below.
Introduction
When authors state” Cytokines involvement in cardiovascular atherosclerotic events is well known, being demonstrated high levels of IL-1, IL-6, TNF-α, IFN-γ, IL-10 and IL-22.” Authors should cite for each cytokine an appropriate reference. This justifies why they measure those cytokines compared to many others involved.
“focus score” should be described and referenced in the abstract and in the Introduction section
Methods
The number of control subject is low compared to patients. The ratio 1:1 or 1: 2 should be better evaluate the differences limiting the risk of bias. Otherwise included power calculation of the study at posteriori.
Endothelial dysfunction was assessed by means of flow-mediated vasodilation (FMD), on brachial artery. No references of this method are provided at all. In addition, authors should include sensitivity and specificity of this method and commented on the studies ‘ results of FMD in primary or (secondary) Sjogren’ syndrome.
Please included details of analytical methods for TNF α, IL-6 and Beta- 2microglobulin: specificity and sensitivity of each technique, limit of detection and if ELISA kits were used included references of the KIT and commercial trademark.
Please include data normality analysis in order to decide if parametric tests should be used. This is crucial and can change the results.
Results
Please analyse gender differences in patients group and control groups.
In Table 2 provided leucocytes’count with their subtypes (platelet and erythrocytes are not necessary but these data will add more clarity to the results of this clinical study).
Please included analysis of leucocytes’ subtypes and main outcomes' variables.
minor spelling
Author Response
Dear Reviewer,
Thank you for the time given to review the manuscript and for the recommendation you have offered.
I have entered the references suggested by you, including those related to cytokines and FMD.
I described the focus score, too.
I added more controls, to have a 1:2 ratio. I introduced the leucocytes’count and lymphocites’ count, too.
The normal distribution of data was assessed using Kolmogorov-Smirnov test (K-S test statistic (D)= 0.06077, p = 0.68231). Normally distributed data were reported as means ± standard deviations. The correlations and differences were assessed with Pearson coefficients and T tests, respectively.
Thank you,
Best regards,
Alexandru Caraba

Reviewer 3 Report
This manuscript investigated the correlation between endothelial dysfunction and serum biomarkers in the primary Sjogren’s syndrome. Although interesting, this manuscript is not well written and must be professionally edited, because such expression reflects the authors’ poor understanding of this study and cannot transfer findings effectively to our readers. Other than the language, this discussion of results is not deep enough to provide some insights; most parts of discussion only provide some pure information without corresponding interpretation from the authors. Since there are too many writing problems, I only provide some of my comments below. For more details, please refer to my attached file as problematic areas are highlighted in your manuscript.
- Line3
The “activity” is not precise. Please change the title for a clear expression.
- Line 15
“Represent” means using something to be a symbol of something. Please indicate why your aim should be “represented” by others. Or the word is misused.
- Line 20
The word “perform” cannot be used here, because serum components cannot be “performed” and only something like experiments can be “performed”.
- Line 21
Please indicate the meaning of “SAA” and “SSB” when they first appear.
- Line 22
The comma in between “t-test, and” should be removed.
- Line 25
Change “at the value of p<0.05” to “if p< 0.05.”
- Line 29
Change “statistically significant” to “with statistical significance”.
- Line 31
There should be an “and” in between “cytokines, beta 2-microglobulin”.
- Line 36
The “, being primary or secondary” is not written.
- Line 37
Remove “:” after “like”.
- Line 39
Change “, but not only” to “etc.”
- Line 41
“, as:” is not right.
- Line 42
Should be “vessels, and central or peripheral nervous system”.
- Line 192 to 199
Discuss your uniqueness when you compare your study with others.
In summary, the manuscript is interesting. However, it needs serious improvement in terms of scientific writing and discussion.

English is not okay.
Author Response
Dear Reviewer,
Thank you for taking the time to review the manuscript and for your reccomendation.
I made the changes you indicated.
The manuscript was read and checked by a colleague, a native English speaker:
Thank you,
Best regards,
Alexandru Caraba

Round 2
Reviewer 2 Report
Authors revised properly
It is fine